# HOME-3: High-Order Momentum Estimator using Third-Power Gradient for Convex, Smooth Nonconvex, and Nonsmooth Nonconvex Optimization

## Abstract

Momentum-based gradients are critical for optimizing advanced machine learning models, as they not only accelerate convergence but also help gradient-based optimizers overcome stationary points. While most state-of-the-art momentum techniques rely on lower-power gradients, such as the squared first-order gradient, there has been limited exploration into the potential of higher-power gradients—those raised to powers greater than two, such as the third-power first-order gradient. In this work, we introduce the concept of high-order momentum, where momentum is constructed using higher-power gradients, with a specific focus on the third-power first-order gradient as a representative example. Our research offers both theoretical and empirical evidence of the benefits of this novel approach. From a theoretical standpoint, we demonstrate that incorporating third-power gradients into momentum can improve the convergence bounds of gradient-based optimizers for both convex and smooth nonconvex problems. To validate these findings, we conducted extensive empirical experiments across convex, smooth nonconvex, and nonsmooth nonconvex optimization tasks. The results consistently showcase that high-order momentum outperforms traditional momentum-based optimizers, providing superior performance and more efficient optimization.

## 1 Introduction

Optimization problems in machine learning are commonly tackled using gradient-based optimizers, which rely on either full gradients—computed from the entire dataset—or stochastic gradients, derived from mini-batches. While full gradients guarantee eventual convergence, stochastic gradients offer enhanced computational efficiency (Hazan et al., 2007; Nemirovski et al., 2009; Rakhlin et al., 2011). Over the past decade, research has shown that combining full gradients, stochastic gradients, noisy stimuli, batch strategies, sampling, and momentum techniques in gradient-based optimizers can lead to favorable convergence, expected accuracy, and improved robustness (Shalev-Shwartz & Zhang, 2013; Zhang et al., 2012; Johnson & Zhang, 2013; Defazio et al., 2014; Arjevani & Shamir, 2015; Lin et al., 2015; Allen-Zhu, 2017; Haji & Abdulazeez, 2021).

Momentum, one of the most influential techniques, is widely used in gradient-based optimizers to further improve performance (Liu et al., 2020; Loizou & Richtárik, 2020; Haji & Abdulazeez, 2021). Intuitively, momentum addresses the issue of slow convergence in later stages of optimization, such as near $(\delta, \epsilon)$-Goldstein stationary points (Clarke, 1974; 1975; 1981; 1990; Jordan et al., 2023), where gradients oscillate within a narrow range. Momentum helps by driving gradients away from these oscillations and toward the global optimum, making it especially effective for nonsmooth nonconvex objectives, such as those found in Deep Neural Networks (DNNs) (Mai & Johansson, 2020; Wang et al., 2021; Wang & Wen, 2022; Jordan et al., 2023).

Due to these advantages, leading optimizers like Adam, STORM, and $STORM^+$ (Kingma & Ba, 2014; Cutkosky & Orabona, 2019; Levy et al., 2021) incorporate momentum to achieve higher accuracy and reduce the likelihood of getting trapped in stationary points. For instance, Adam uses two momentum terms—first-order and squared first-order gradients—to optimize objective functions,

often outperforming alternatives like AdaGrad and SGD (Kingma & Ba, 2014; Lydia & Francis, 2019; Chandra et al., 2022; Beznosikov et al., 2023). STORM, which uses a stochastic recursive momentum term based on squared gradients, has been shown to achieve better accuracy than Adam when optimizing ResNet (Cutkosky & Orabona, 2019), and the more recent $STORM^+$ enhances this approach with adaptive learning rates, eliminating the need for parameter tuning (Levy et al., 2021).

While first-order and squared gradients dominate current momentum-based approaches, exploring higher-order momentum holds great potential. For instance, incorporating third-power gradients could further enhance the convergence bound of gradient-based optimizers. In this work, we introduce the High-Order Momentum Estimator (HOME) optimizer, a framework designed to explore and advance high-order momentum techniques. Our focus is on *HOME*-3, which leverages third-power gradients to enhance momentum, such as $(f')^3$. First, we present a theoretical analysis showing that *HOME*-3 significantly improves convergence bounds for both convex and smooth nonconvex optimization problems. We then extend our numerical experiments to nonsmooth nonconvex problems, where *HOME*-3 consistently outperforms other momentum-based optimizers. Finally, we use statistical techniques to quantify the performance of *HOME*-3, validating both the effectiveness and robustness of third-power gradients in momentum.

**Contributions**: In this work, the potential contributions of *HOME* are categorized as follows:

*Third-Order Momentum Enhances Convergence Bound for Convex Problems* (**Theorem** 4.1): Based on the assumptions and properties of convex objective functions (see **Assumption** 2.1), the proposed *HOME*-3 optimizer, incorporating a third-power gradient, enhances the convergence bound to $O(1/T^{5/6})$. Detailed proof of **Theorem** 4.1 can be found in Appendix A of the Supplementary Material.

*Third-Order Momentum Advances Convergence Bound for Smooth Nonconvex Problems* (**Theorem** 4.2): According to the assumptions and properties of smooth nonconvex functions (see **Assumption** 2.2), the *HOME*-3 optimizer advances the convergence bound to approximately $O(1/T^{5/6})$. The proof for **Theorem** 4.2 is provided in Appendix A of the Supplementary Material.

*High-Order Momentum Enhances Convergence for Nonsmooth Nonconvex Problems* (**Theorem** 4.4): We empirically investigate the performance of high-order momentum optimizers on nonsmooth nonconvex problems, as illustrated in Figure 3 . To further validate the performance of *HOME*-3, we employ a deep neural network, since the objective function of a multi-layer deep neural network is typically nonsmooth and nonconvex (Jordan et al., 2023). The results, shown in Figures 3 and 4, indicate that *HOME*-3 outperforms other peer momentum-based optimizers. Additionally, we explore the advantages of coordinate randomization in **Lemma** 4.3 and **Theorem** 4.4, demonstrating that it preserves the convergence bound of the original gradient-based optimizer.

**Related Work**: In the field of convex and smooth nonconvex optimization, Kingma's work on Adam (Kingma & Ba, 2014) demonstrated that momentum, built on the first-order and squared gradients, can achieve a convergence bound of $O(1/T^{1/2})$ for convex problems. Similarly, STORM, which uses a recursive stochastic momentum, obtains a convergence bound of $O(1/T^{1/3})$ for smooth nonconvex problems (Cutkosky & Orabona, 2019). More recently, $STORM^+$ achieved a bound of $O(1/T^{1/2} + \sigma^{1/3}/T^{1/3})$ (Levy et al., 2021). Notably, in both convex and smooth nonconvex scenarios, *HOME*-3 achieves a superior convergence bound of $O(1/T^{5/6})$.

## 2 PRELIMINARIES: DEFINITIONS AND ASSUMPTIONS

In this work, we consider the following minimization problem:

$$\min_{X \subseteq \mathbb{R}^D} f(X) \tag{1}$$

In equation 1, $X \in \mathbb{R}^D$ represents a vector denotes the independent variables of an objective function $f(\cdot) : \mathbb{R}^D \to \mathbb{R}, D < \infty$. The objective function $f(\cdot)$ can be convex, smooth nonconvex, or nonsmooth nonconvex real functions. In this work, the theoretical analyses of the convergence bound concentrate on convex and smooth nonconvex problems. Due to the advantages of coordinate randomization on nonsmooth nonconvex problems that have been discussed recently, we empirically investigate the performance of momentum incorporating coordinate randomization and third-power

gradient on nonsmooth nonconvex optimization (Zhang & Bao, 2022). Furthermore, we have essential definitions and assumptions throughout this work as follows:

**Definition 2.1** *(High-Order Momentum) Given a momentum $M$ denoted on $x \in \mathbb{R}^D$, $M$ relies on variables $\{\nabla f(x), (\nabla f(x))^2, \cdots, (\nabla f(x))^n\}$, $n < \infty$, we call $M$ a $n^{th}$-order momentum. And $n$ is the maximum power of the gradient employed to build the momentum.*

**Definition 2.2** *(Smooth Property) Given an objective function $f$ denoted as $f : \mathbb{R}^D \to \mathbb{R}$, for any $k \in \mathbb{N}$, if $\left\| \nabla^k f(x) - \nabla^k f(y) \right\| \leq L \left\| x - y \right\|$ holds, we call $f$ a smooth function. And $\|\cdot\|$ represents an Euclidean norm. We can denote $g = \nabla f(x)$ and $g_t = \nabla f(x_t)$. $g_t$ represents a gradient within $t$ iterations.*

**Definition 2.3** *(Gradient-based Optimization Operator) Given an operator as $\mathcal{G} : \mathbb{R} \to \mathbb{R}^D$, $\mathcal{G}$ denotes a gradient-based optimization operator. For example, suppose $t(\forall t \in \mathbb{N})$ as current iteration, we have $\mathcal{G} \cdot f(x_t) = x_t - \alpha \cdot \nabla f(x_t)$, the operator $\mathcal{G}$ denotes a first-order gradient-based optimizer.*

**Definition 2.4** *(Coordinate Randomization) Given an operator $\mathcal{R}$ denoted as $\mathcal{R} : \mathbb{R}^D \to \mathbb{R}^D$, we have $\mathcal{R}[x_1, x_2, \cdots, x_D] \to [\hat{x}_1, \hat{x}_2, \cdots, \hat{x}_D]$. The operator $\mathcal{R}$ is a coordinate randomization.*

**Definition 2.5** *(Iterative Format of Gradient and Permutation Randomization Operators) Given gradient and permutation randomization operators $\mathcal{G}$ and $\mathcal{R}$, suppose the current iteration as $t$, $\mathcal{G}^t f(x)$ and $\mathcal{R}^t x$ represent an iterative format of gradient and permutation randomization operator within $t$ iterations. For example, $\mathcal{G}^2 f(x) = \mathcal{G} \cdot \mathcal{G} \cdot f(x)$ and $\mathcal{R}^2 x = \mathcal{R} \cdot \mathcal{R} \cdot x$.*

**Definition 2.6** *(Initialization and Stationary Point) We denote $x_0$ as an initialized variable for a gradient-based optimizer to begin iteration. Meanwhile, a stationary point is represented by $x_T$, and $T$ indicates the maximum iteration.*

**Definition 2.7** *(Iterative Output of Gradient and Coordinate Randomization Operators) Given gradient and permutation randomization operators $\mathcal{G}$ and $\mathcal{R}$, suppose the current iteration as $t$, $\mathcal{G}^t f(x)$ and $\mathcal{R}^t x$ represent gradient and permutation randomization operator within $t$ iterations. The iterative output of gradient and permutation randomization operators are denoted as $x_t = \mathcal{G} \cdot f(x_{t-1}) = \mathcal{G}^t \cdot f(x_0)$ and $\hat{x}_t = \mathcal{R} \cdot \mathcal{G} \cdot f(x_{t-1}) = \mathcal{R}^t \cdot \mathcal{G}^t \cdot f(x_0)$.*

Moreover, four vital assumptions are provided below to benefit theoretical analyses of *HOME*-3 optimizer on convex, smooth nonconvex, and nonsmooth nonconvex optimization.

**Assumption 2.1** *(Convex Assumption) $f(y) \geq f(x) + (\nabla f(x))^T (y - x)$, $x, y \in \mathbb{R}^D$*

**Assumption 2.2** *(Smooth Nonconvex Assumption) $f(y) \leq f(x) + (\nabla f(x))^T (y - x) + \frac{L}{2} \cdot \|x - y\|$, $x, y \in \mathbb{R}^D, L \in \mathbb{R}, L > 0$*

**Assumption 2.3** *(Finite Dimensional Assumption) In this study, the objective function $f : \mathbb{R}^D \to \mathbb{R}$, gradient optimizer $\mathcal{G}$ denoted as $\mathcal{G} : \mathbb{R} \to \mathbb{R}^D$, and coordinate randomization $\mathcal{R} : \mathbb{R}^D \to \mathbb{R}^D$, all theoretical analyses under $D < \infty$.*

**Assumption 2.4** *(Linear Representation of All Gradients) Considering iteration from $1$ to $T$, for any $t \in [1, T]$, and $n \in \mathbb{N}$ as the power for gradient, $\forall \epsilon > 0$, the following equation holds:*

$$\|g_t^n - (k_1 g_1^n + k_2 g_2^n + \cdots + k_T g_T^n)\| < \epsilon \tag{2}$$

*$\{k_1, k_2, \cdots, k_T\}$ are constant and $\{g_1, g_2, \cdots, g_T\}$ represents first-order gradient in $1, 2, \cdots, T$ iteration.*

We can derive equation 2 from the **Assumption** 2.2 when the objective function is smooth. In fact, if and only if $\forall i, j \in \mathbb{N}, i \neq j, corr(g_i, g_j) = 0$, equation 2 holds.

## 3 METHOD: HIGH-ORDER MOMENTUM ESTIMATOR (*HOME*)

This section outlines the details of the *HOME* optimizer, as summarized in Table 1. At its core, the *HOME* optimizer offers a framework for incorporating high-power first-order gradients to generate high-order momentum.

In particular, we focus on analyzing the properties of high-order momentum using a third-power first-order gradient as a starting point and extend our theoretical analysis to even higher-order momenta, such as those utilizing a sixth-power gradient. To facilitate both implementation and validation against other state-of-the-art optimizers, we base our framework on the widely used Adam optimizer. However, in contrast to Adam, which is dominated by first- and second-order momentum terms, our proposed method introduces an innovative update rule that is driven by the interaction between the first and third momentum terms, as shown below:

$$x_t \leftarrow x_{t-1} - \alpha_t \cdot (\hat{M}_{t-1} - \hat{S}_{t-1})/(\sqrt{\hat{V}_{t-1}} + \epsilon_1) \tag{3}$$

In equation 3, $\hat{M}_t$, $\hat{V}_t$, and $\hat{S}_t$ denote the first-order, second-order, and third-order momentum (please refer to **Definition** 2.1). Meanwhile, $\alpha_t$ denotes an adaptive learning rate (Huang et al., 2021). And $\epsilon_1$ is set the same as Adam (Kingma & Ba, 2014). In addition, the third momentum term $\hat{S}_t$ is cultivated on the third-power first-order gradient:

$$\begin{aligned} S_t &\leftarrow \beta_3 S_{t-1} + (1 - \beta_3)g_t^3 \\ \hat{S}_t &\leftarrow \frac{S_t}{1 - \beta_3^t} \end{aligned} \tag{4}$$

where $\beta_3$ is an exponential decay and $g_t^3$ represents a third-power gradient within iteration $t$. Intuitively, a higher-power gradient dominates the update when the gradient norm is sufficiently large at the early stage. Otherwise, a lower-order gradient is in charge of the update when the gradient norm is reduced to a small value. That is, the convergence bound of the *HOME* optimizer is adaptive. In addition, other efficient techniques are included for the *HOME* optimizer, such as adaptive learning rate (Huang et al., 2021) and coordinate randomization (Zhang & Bao, 2022) since these techniques guarantee an influential impact (Huang et al., 2021; Jordan et al., 2023) on complex optimization, e.g., nonsmooth/smooth nonconvex problems.

The input for *HOME*-3 optimizer is: $t$ represents current iteration; $T$ defines the maximum iteration; $\alpha_t$ denotes an adaptive step size based on current iteration (Huang et al., 2021), such as $0.001 \times (1 - \frac{t}{T})$; $\beta_1 = 0.9$, $\beta_2 = 0.999$, $\beta_3 = 0.99$ are exponential decay for three momentum terms (Kingma & Ba, 2014), respectively; Notably, $\beta_3$ is manually set, ensuring that $\beta_1 < \beta_3 < \beta_2$; $M_0$ denotes the first-moment vector and initializes as 0; $V_0$ denotes the second momentum vector and is initialized as 0; $S_0$ denotes the third momentum vector and is initialized as 0; $\epsilon_1$ defines the same in Adam; $\epsilon_2$ represents a threshold when gradient within a stationary point. In this work, we set $\epsilon_2$ the same as $\epsilon_1$.

Importantly, Table 1 presents a framework updated on Adam optimizer (Kingma & Ba, 2014) to introduce one additional momentum term using a third-power gradient to improve the convergence bound. The *HOME*-3 indicates that the highest power of the gradient for cultivating momentum is 3. Notably, the coordinate randomization $\mathcal{R}$ is only applied to nonsmooth nonconvex problems. Thus, the framework in Table 1 could be treated as a potential standard framework to incorporate high-order momentum.

As discussed before, a higher-order momentum $S_t$ and $\hat{S}_t$ dominate the update at the beginning, due to $\|g_t^3\| >> \|g_t\|$. Furthermore, when the gradient approximates a stationary point or local optimum, such as $\forall \epsilon > 0$, $\|g_t\| < \epsilon$, the lower-power gradient is in charge of updating. In particular, let the Eq. 3 equal to 0, we can infer the stopping criteria of *HOME*-3:

$$\forall \epsilon > 0 \left\| \hat{M}_t - \hat{S}_t \right\| < \epsilon \tag{5}$$

Since $\left\| \hat{M}_t - \hat{S}_t \right\| < \epsilon$ can result in terminating *HOME*-3, as indicated in equation 4 and equation 5, we introduce coordinate randomization for *HOME* optimizers to escape potential stationary points in the objective function. Furthermore, at the late stage, when the gradient approximates to the stationary point, such as $\left\| \hat{M}_t \right\|$, $\left\| \hat{S}_t \right\| < \epsilon$, coordinate randomization can maintain the difference between $\left\| \hat{M}_t \right\|$ and $\left\| \hat{S}_t \right\|$ in order to advance $\hat{S}_t - \hat{M}_t$ to escape an open cube of stationary points.

Table 1: The Pseudo Code of High-Order Momentum Estimator (HOME)

| **Algorithm 1:** *HOME*-3 |
| --- |
| 1: **while** $t < T$ |
| 2: $\quad g_t \leftarrow \nabla_x f(x_t)$ |
| 3: $\quad M_t \leftarrow \beta_1 M_{t-1} + (1 - \beta_1) g_t$ |
| 4: $\quad V_t \leftarrow \beta_2 V_{t-1} + (1 - \beta_2) g_t^2$ |
| 5: $\quad S_t \leftarrow \beta_3 S_{t-1} + (1 - \beta_3) g_t^3$ |
| 6: $\quad \hat{M}_t \leftarrow \frac{M_t}{1 - \beta_1^t}$ |
| 7: $\quad \hat{V}_t \leftarrow \frac{V_t}{1 - \beta_2^t}$ |
| 8: $\quad \hat{S}_t \leftarrow \frac{S_t}{1 - \beta_3^t}$ |
| 9: $\quad x_{t+1} \leftarrow x_t - \alpha_t \cdot (\hat{M}_t - \hat{S}_t)/(\sqrt{\hat{V}_t} + \epsilon_1)$ |
| 10: $\quad$ **if** $\left\| \hat{M}_t - \hat{S}_t \right\| < \epsilon_2$ |
| 11: $\quad\quad \hat{x}_{t+1} \leftarrow \mathcal{R}(x_{t+1})$ |
| 12: $\quad\quad x_{t+1} \leftarrow \hat{x}_{t+1}$ |
| 13: $\quad$ **End if** |
| 14: $\quad t \leftarrow t + 1$ |
| 15: **End while** |

# 4 THEORETICAL ANALYSES

This section presents the convergence analyses of the *HOME*-3 optimizer under three assumptions. We begin by examining the convex case that satisfies Assumption 2.1, demonstrating that *HOME*-3 can achieve a convergence upper bound of $O(1/T^{5/6})$, as outlined in Section 4.1. In Section 4.2, we extend this analysis under Assumption 2.2, showing that the convergence bound of the *HOME*-3 optimizer remains comparable to that of the convex case. Additionally, in Section 4.3, we introduce a key advancement—coordinate randomization—which can further enhance the performance of *HOME*-3 in nonsmooth nonconvex scenarios. The results partially answer the questions *What is the role of randomization in dimension-free nonsmooth nonconvex optimization* raised by Jordan (Jordan et al., 2023). In short, complete theoretical proofs for the *HOME*-3 optimizer are provided in Appendix A of the Supplementary Material.

## 4.1 CONVEX CASE

We theoretically analyze the convergence bound of *HOME*-3 under the convexity assumption (please refer to **Assumption** 2.1) in this section. The following **Theorem** 4.1 demonstrates *HOME*-3 can reach a convergence bound as $O(1/T^{5/6})$.

**Theorem 4.1** *Let $f$ satisfy **Assumption 1**, suppose $T$ as the maximum iteration, inferring from **Definitions** 2.3, 2.5, and 2.6, then $\frac{\left\| \Sigma_{t=1}^T (f(x_t) - f(x_T)) \right\|}{T}$ can reach $O(1/T^{1/6})$.*

The detailed proof of **Theorem** 4.1 can be viewed in Appendix A, Supplementary Material.

## 4.2 SMOOTH NONCONVEX CASE

In this section, under the smooth nonconvex Assumption (please refer to **Assumption** 2.2, we prove that the convergence bound of *HOME*-3 can approximately reach $O(1/T^{1/6})$. The potential issue impacting the convergence bound of *HOME*-3 is the term $\frac{L}{2} \cdot \|x - y\|$. According to our analyses, if $T$ is sufficiently large and guarantees $\frac{L}{\sqrt{T}} \to 0, \forall x, y \in X$, in that case, the convergence bound of *HOME*-3 is comparable to convexity assumption (please refer to **Assumption** 2.1). Similarly, the convergence upper bound under smooth nonconvex cases can reach to $O(1/T^{1/6})$.

**Theorem 4.2** *Let $f$ satisfy **Assumption 2**, suppose $T$ as the maximum iteration, inferring from **Definitions** 3, 5, and 6, then $\frac{\|f(x_t) - f(x_T)\|}{T}$ can reach $O(1/T^{5/6})$.*

The detailed proof of **Theorem** 4.2 can be viewed in Appendix A, Supplementary Material.

### 4.3 NONSMOOTH NONCONVEX CASE

Due to the complexity of smooth nonconvex cases, $\left\|\hat{M}_t - \hat{S}_t\right\|$ could be 0 when the gradient approximates the stationary point. To overcome this challenge, we incorporate randomization to increase the opportunity for the optimizer to approximate an open cube of the global optimum. Notably, the following Lemma proves that the norm of coordinate randomization is equal to 1.

**Lemma 4.3** *(Norm of Coordinate Randomization Operator is Equal to 1) Suppose the permutation randomization as an operator $\mathcal{R} : \mathbb{R}^D \to \mathbb{R}^D$, $\|\mathcal{R}\| = 1$ holds, if $D < \infty$.*

It is not difficult to prove **Lemma** 4.3. The proof of Lemma 4.3 can be viewed in Appendix A, Supplementary Material.

Importantly, in **Theorem** 4.4, we discuss the upper bound on the convergence bound of gradient-based optimizer (Wang & Shen, 2023) incorporating coordinate randomization is comparable to $\left\|\mathcal{G}^{t+1} \cdot f(x_0) - \mathcal{G}^t \cdot f(x_0))\right\|$; thus, we discuss that coordinate randomization could maintain the convergence bound of incorporated gradient-based optimizer and is shown in **Theorem** 4.4.

According to Definition 2.3, we can infer:

$$\|\mathcal{R} \cdot [x_1, x_2, \cdots, x_D]\| = \|[\hat{x}_1, \hat{x}_2, \cdots, \hat{x}_D]\| \tag{6}$$

According to **Definition** 2.5, **Lemma** 4.3, and **Assumption** 2.3, for any $x, y \in I$, we have:

$$\left\|\mathcal{R}^t \cdot \mathcal{G}^t \cdot (f(x) - f(y))\right\| \leq \left\|\mathcal{R}^t\right\| \cdot \left\|\mathcal{G}^t \cdot (f(x) - f(y))\right\| = \left\|\mathcal{G}^t \cdot (f(x) - f(y))\right\| \tag{7}$$

Let $x$ be $x_1 = \mathcal{G} \cdot f(x_0)$ and $Y$ be $x_0$, inferring from equation 5, we have:

$$\left\|\mathcal{R}^t \cdot \mathcal{G}^t \cdot (f(x_1) - f(x_0))\right\| \leq \left\|\mathcal{G}^{t+1} \cdot f(x_0) - \mathcal{G}^t \cdot f(x_0))\right\| \tag{8}$$

**Theorem 4.4** *(Coordinate Randomization Maintains The Convergence Bound of Incorporated Optimizer) Inferring from **Lemma 4.3**, the convergence bound of a gradient-based optimizer incorporating coordinate randomization $\mathcal{R} \cdot \mathcal{G}$ should be equal to the convergence bound of an original gradient-based optimizer $\mathcal{G}$ without coordinate randomization.*

## 5 NUMERICAL EXPERIMENTS

We validate *HOME* with three other peer optimizers, such as ADMM (Nishihara et al., 2015), Adam (Kingma & Ba, 2014), and STORM (Cutkosky & Orabona, 2019), on the public biomedical data in Multiband Multi-echo (MBME) functional Magnetic Resonance Imaging (fMRI) (Wang, 2018). After pre-processing (Ji et al., 2022), the size of each input signal matrix is $100 \times 902,629$. The total number of subjects is 29. In this empirical study, all optimizers are terminated after 100 iterations with other parameters fixed to the reported default values in the literature (Kingma & Ba, 2014; Cutkosky & Orabona, 2019; Nishihara et al., 2015). In addition, $\epsilon_2$ representing the difference between the previous and current gradient is the same as $\epsilon_1$ (Kingma & Ba, 2014). Furthermore, the experimental studies are validated on the CPU cluster, including 16 Intel Xeon X5570 2.93GHz. Moreover, to facilitate statistical analyses based on a large number of augmented subjects, the original 29 subjects are expanded to 100 via data augmentation techniques (Wen et al., 2020; Iwana & Uchida, 2021).

### 5.1 EXPERIMENT ON CONVEX PROBLEM: DICTIONARY LEARNING

Since Dictionary Learning (DL) is one of the representative alternative convex problems (Hao et al., 2023; Tošić & Frossard, 2011), we employ *HOME*-3 and other peer optimizers to optimize the objective functions of DL presented as follows:

$$\min_{X,Y \in \mathbb{R}^{p \times q}} \|I - XY\| + \lambda \|Y\|_1, \, p, q \in \mathbb{N} \tag{9}$$

In equation 9, $I$ denotes the input matrix. $X$ and $Y$ denote weight and feature matrices, respectively. $\lambda$ represents a sparse trade-off set as the default value (Tošić & Frossard, 2011). Since DL is an alternative convex problem, we can validate the theoretical conclusion in Section 4.1. In addition, we provide a reconstruction loss to compare *HOME* with other peer optimizers quantitatively. And, since DL is an unsupervised learning problem, we provide the reconstruction loss in Eq. 10 as follows:

$$Reconstruction\ Loss = \frac{\|I - XY\|}{\|I\|} \tag{10}$$

Overall, Figure 1 presents the averaged reconstruction loss of *HOME*-3 and other peer optimizers to optimize the objective function of DL. In particular, according to Figure 1 (a), *HOME*-3 can enhance the convergence and reconstruction accuracy. Notably, *HOME*-3 demonstrates a more extensive reconstruction loss at the early stage due to a larger norm of high-power gradient. In Figure 1 (b), in this most straightforward case, an individual reconstruction loss reveals the convergence of ADMM (Nishihara et al., 2015) is faster than Adam (Kingma & Ba, 2014) and STORM (Cutkosky & Orabona, 2019) but *HOME*-3 obtains the steepest convergence curve at the early stage.

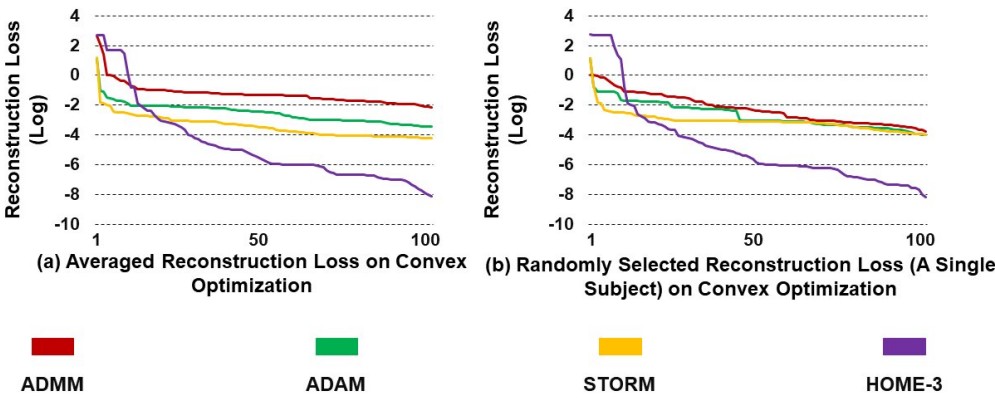

Figure 1: Averaged reconstruction loss comparison of proposed *HOME*-3 and other three peer optimizers within one hundred iterations

## 5.2 EXPERIMENT ON SMOOTH NONCONVEX PROBLEM: DEEP NONLINEAR MATRIX FACTORIZATIONS

Furthermore, to validate *HOME*-3 on smooth nonconvex optimization, we introduce the objective functions of Deep Nonlinear Matrix Factorization (DNMF) (Trigeorgis et al., 2016), presented in equation 11a and equation 11b. Overall, DNMF is comparable to layer-stack deep neural networks such as a Deep Belief Network (DBN) consisting of multiple restricted Boltzmann machines (Hinton, 2009; Gu et al., 2022). Meanwhile, similar to DBN, since DNMF is an unsupervised learning problem, we focus on comparing reconstruction loss in the following Figure 2. Importantly, to avoid arbitrary hyperparameter tuning, we employ a rank estimator (Zhao & Zhao, 2020) to automatically estimate the number of layers and layer size. For activation function between adjacent layers, considering previous works (Jordan et al., 2023), we set Rectified Linear Unit (ReLU) (Agarap, 2018) as an activation function $\mathcal{N}_k$ in equation 11b to increase the complexity of objective function in DNMF.

$$\min_{Z_i \in \mathbb{R}^{p \times q}} \bigcup_{i=1}^{k} \|Z_i\|_1 \tag{11a}$$

$$s.t. (\prod_{i=1}^{k} X_i) \cdot \mathcal{N}_k(Y_k) + Z_k = I \tag{11b}$$

In equation 11, $I$ denotes the input matrix. $X_i$ denotes the current layer and $Y_i$ denotes the current feature matrix. In addition, $\mathcal{N}_k$ represents an activation function in the current layer. Lastly, $Z_k$ indicates a background noise matrix. And $k$ represents the total layer number.

In addition, reconstruction loss under smooth nonconvex assumption is denoted as:

$$Reconstruction\ Loss = \frac{\left\| \left( \prod_{i=1}^{k} X_i \right) \cdot \mathcal{N}_k(Y_k) + Z_k - I \right\|}{\|I\|} \tag{12}$$

In the following Figure 2, we present a reconstruction loss to compare the *HOME*-3 with other peer optimizers in the first and second layers of DNMF. Overall, in Figure 2 (a) and (b), *HOME*-3 has improved the convergence. Even in the late stage (after 60 iterations), due to the high-order momentum, *HOME*-3 can still converge faster than peer optimizers.

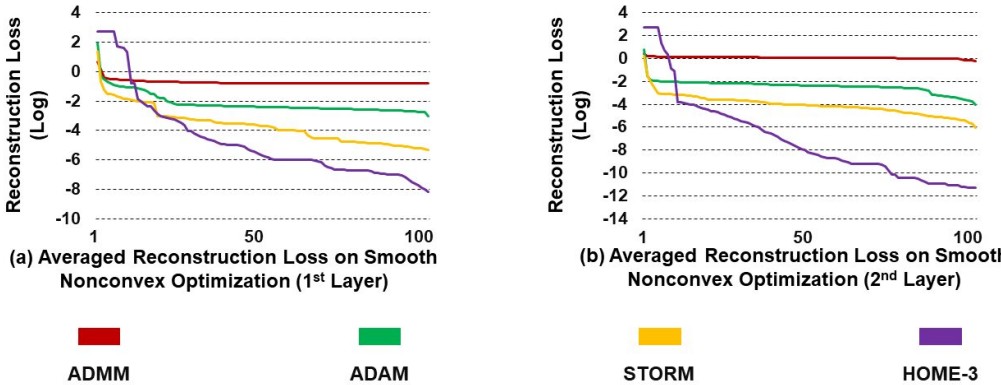

Figure 2: Averaged reconstruction loss comparison of proposed *HOME*-3 and other three peer optimizers with in one hundred iterations at first and second layers of DNMF

### 5.3 EXPERIMENT ON NONSMOOTH NONCONVEX PROBLEM: NOISY DEEP MATRIX FACTORIZATION

Moreover, in this section, to continuously increase the complexity in objective functions, we aim to investigate the performance of *HOME*-3 optimizer under the nonsmooth nonconvex case. To implement a nonsmooth nonconvex optimization, we add additional random noise to the feature matrix in DNMF (Lu et al., 2014; Lin et al., 2022), such as:

$$Y_i \leftarrow Y_i + random\ noise \tag{13}$$

In equation 13, a random noise is added to the feature matrix $Y_i$ in equation 11. The random noise results in nonsmooth nonconvex objective functions (Lu et al., 2014; Lin et al., 2022). Importantly, to avoid the noise overwhelming the original data, we set the boundary of random noise in this experiment as $[-0.1 \cdot Median, 0.1 \cdot Median]$. $Median$ represents the median of the input matrix or vector.

Figure 3 compares reconstruction loss of *HOME*-3 with other peer optimizers under the nonsmooth nonconvex case. Even in the most complex case, *HOME*-3 can still enhance the convergence and provide most accurate reconstruction. In Figures 3 (a) and (b), it is noticeable that the convergence curve of *HOME*-3 is steepest within 20 iterations. The results further demonstrate that the high-order momentum can improve the convergence and maintain the impact until the late stage (please refer to the convergence curve in Figure 3 (a) and (b) after 80 iterations). Meanwhile, empirical results suggest that the use of coordinate randomization can benefit gradient optimizers by increasing reconstruction accuracy.

### 5.4 EXPERIMENT ON NONSMOOTH NONCONVEX PROBLEM: DEEP NEURAL NETWORK

Besides, Figure 4 presents a comparison between *HOME*-3 and two other leading optimizers—ADAM (Kingma & Ba, 2014)and STORM (Cutkosky & Orabona, 2019)—in optimizing a three-layer DBN Hinton (2009). The DBN uses ReLU (Agarap, 2018), and the reconstruction loss

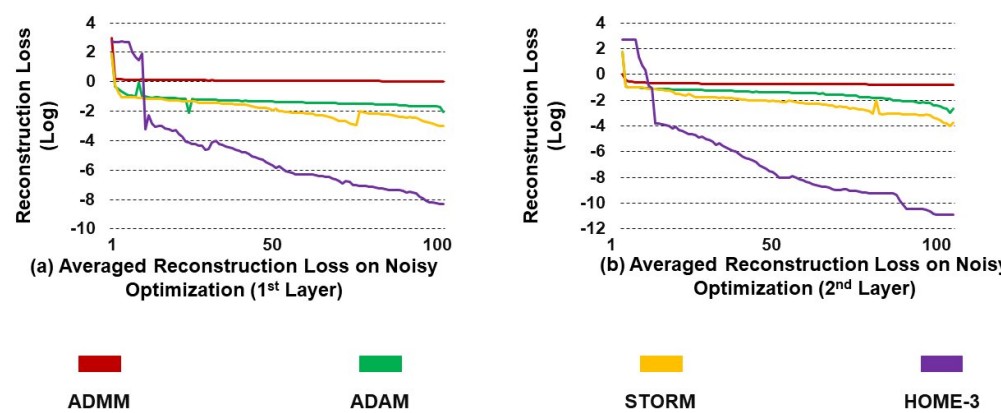

Figure 3: The averaged training loss comparison of proposed *HOME*-3 and other three peer optimizers within one hundred iterations of all subjects at first and second layers of noisy DNMF, respectively.

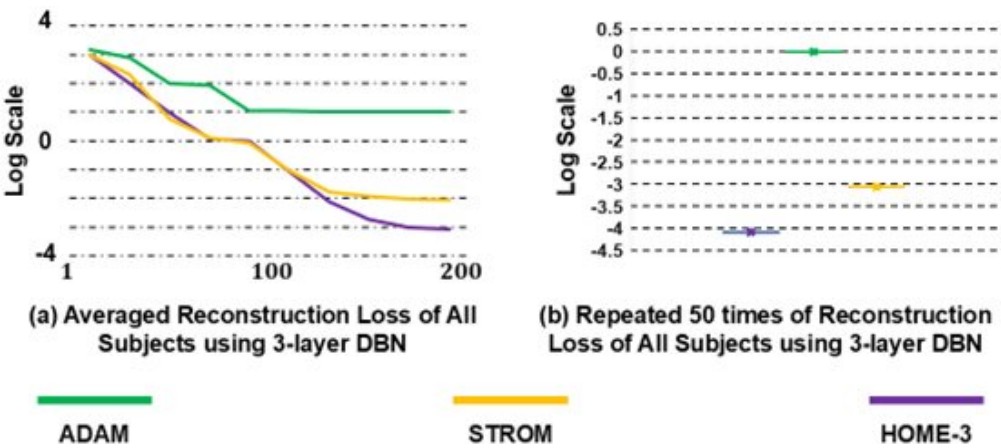

Figure 4: An illustration of reconstruction loss comparisons of *HOME*-3 and other peer optimizers on optimizing 3-layer DBN.

follows the same definition as in equation 12. Notably, *HOME*-3 achieves the highest reconstruction accuracy among the methods compared.

Lastly, consistent with previous numerical experiments, *HOME*-3 is validated on supervised learning problems (e.g., Logistic Regression (Schober & Vetter, 2021) using publicly released breast cancer data (Shut, 2023) with other peer optimizers. For a more detailed presentation of these results, please refer to Figures 6 in Appendix A of the Supplementary Materials.

## 5.5 STATISTICAL ANALYSES

In this section, we quantitatively analyze previous experimental results on a large number of samples. Due to all gradient-based optimizers in the empirical study being iterative algorithms, iterative reconstruction loss (please refer to Figures 1, 2, and 3) within each adjacent iteration is not independent. The non-independency limits to directly employ a *t-test* and/or confidential intervals to compare all iterative reconstruction accuracy (Field, 2013). Alternatively, Intra-class correlation coefficients (ICCs), a descriptive statistic technique that can be used for quantitative measurements organized into groups (Bujang & Baharum, 2017).

In Figures 5 (a), (b), and (c), we report the ICCs of *HOME*-3 and three other peer optimizers on previous empirical experiments in Sections 5.1, 5.2, and 5.3. In particular, Figure 5 (a) describes the ICCs on reconstruction loss of *HOME*-3, ADMM (Nishihara et al., 2015), Adam (Kingma & Ba, 2014), and STORM (Cutkosky & Orabona, 2019) on 100 subjects. ADMM is the most robust on convex optimization, and *HOME*-3 is more robust than Adam and STORM (Kingma & Ba, 2014; Cutkosky & Orabona, 2019). In addition, Figure 5 (b) presents the robustness of *HOME*-3, ADMM (Nishihara et al., 2015), Adam (Kingma & Ba, 2014), and STORM (Cutkosky & Orabona, 2019) on smooth nonconvex optimization using 100 subjects. In particular, *HOME*-3 achieves the most robust reconstruction accuracy since the ICCs in both the first and second layers are close to 0.93 and 0.95. Although ADMM obtains the largest ICCs, its reconstruction loss is inaccurate in Figure 2. Notably, though coordinate randomization is introduced, *HOME*-3 is more consistent than Adam and STORM on smooth nonconvex optimization. Lastly, in Figure 5 (c), the robustness of *HOME*-3 is higher than Adam and STORM. There is no significant difference between the first and second layers using *HOME*-3 to optimize nonsmooth nonconvex deep models.

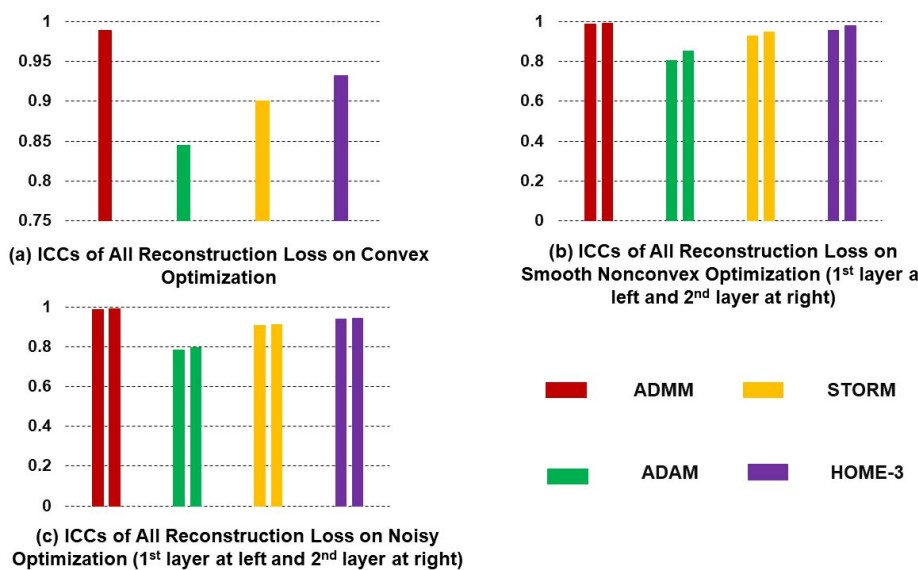

Figure 5: Consistency and robustness comparisons of the proposed *HOME*-3 and three peer algorithms are presented. In Figure 5, panels (a) and (b) demonstrate the ICC values for all optimizers across subjects on convex and smooth nonconvex optimization, respectively. Additionally, Figure 5(c) provides the ICC values that further indicate the consistency and robustness of the *HOME*-3 optimizers.

## 6    CONCLUSION

This work introduces an innovative high-order momentum technique that utilizes high-power gradients to significantly enhance the performance of the gradient-based optimizer. Our contributions are both theoretical and empirical. On the theoretical side, we demonstrate that high-order momentum with high-power gradients improves the convergence bound of optimizers in both convex and smooth nonconvex cases, achieving an upper bound of $O(1/T^{5/6})$. Empirically, extensive experiments showcase that *HOME*-3 consistently delivers superior reconstruction accuracy across convex, smooth nonconvex, and nonsmooth nonconvex problems, underscoring its robustness. Looking ahead, an exciting direction for future research is determining the optimal order of momentum for complex objective functions, which will be pivotal in efficiently optimizing Large Language Models.

ACKNOWLEDGMENTS

We sincerely appreciate the effort of the ICLR 2025 reviewers in improving the quality of our work.

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

## A  APPENDIX

**Proofs**:

**Theorem 4.1** Let $f$ satisfy *Assumption* 1, suppose $T$ as the maximum iteration, inferring from *Definitions* 3, 5, and 6, then $\frac{\left\|\Sigma_{t=1}^{T}(f(x_t)-f(x_T))\right\|}{T} = O(1/T^{5/6})$ holds.

***Proof***: According to **Theorem** 10.5 in Kingma's work Kingma & Ba (2014) and Theorem 4 in Reddi's work Reddi et al. (2019) , suppose the current iteration is $t$, we have the iterative format of *HOME*-3 as:

$$x_{t+1} = x_t - \alpha \cdot \frac{\hat{M}_t - \hat{S}_t}{\sqrt{\hat{V}_t}} \tag{A1}$$

Then, we subtract scalar $x_T$ and square the both side of equation A1,

$$(x_{t+1} - x_T)^2 = (x_t - x_T)^2 - 2\alpha \cdot \frac{(\hat{M}_t - \hat{S}_t)}{\sqrt{\hat{V}_t}} \cdot (x_t - x_T) + \alpha^2 \cdot (\frac{\hat{M}_t - \hat{S}_t}{\sqrt{\hat{V}_t}})^2 \tag{A2}$$

Inferring from equation A2, due to initial value $\hat{S}_0$ equal to 0, $\hat{S}_t$ can be considered a linear combination of cubed gradient $g_t^3$:

$$\hat{S}_t = k_1 \cdot g_1^3 + k_2 \cdot g_2^3 + \cdots + k_t \cdot g_t^3 \tag{A3}$$

In equation A3, $\{k_i\}_{i=1}^t$ is coefficient only relating to $\beta_3$.

Next, inferring from **Definition** 2.3, $\hat{S}_t$ is bounded. We have:

$$\left\|\hat{S}_t\right\| \leq max(\left\|\{k_i\}_{i=1}^t\right\|) \cdot max(\left\|\{g_t\}_{t=1}^T\right\|) \tag{A4}$$

Similarly, inferring from equation A4, we can prove that the first and second momentum, $\hat{M}_t$ and $\hat{V}_t$, are also bounded. Hereby, according to equation A4, we categorize the convergence bound under convexity into two folds:

1). When $g_t$ is sufficiently large, for example $\|g_t\| > 1$, we have $\left\|g_t^3\right\| >> \|g_t\|$. Thus, when $g_t$ is sufficiently large to conveniently analyze the convergence bound, we can ignore the influence from $\hat{M}_t$. In that case, inferring from equation A4, we have:

$$(x_{t+1} - x_T)^2 = (x_t - x_T)^2 + 2\frac{\alpha}{\sqrt{\hat{V}_t}}(\beta_3 S_{t-1} + (1-\beta_3)g_t^3)(x_t - x_T) + \alpha^2 \frac{\hat{S}^2}{\hat{V}_t} \tag{A5}$$

We can infer from equation A5:

$$g_t^3(x_T - x_t) = \frac{\sqrt{\hat{V}_t}}{2\alpha_t(1-\beta_3)}[(x_t - x_T)^2 - (x_{t+1} - x_T)^2] + \frac{\beta_3}{1-\beta_3}S_{t-1} + \frac{\alpha_t}{1-\beta_3} \cdot \frac{\hat{S}^2}{\sqrt{\hat{V}_t}} \tag{A6}$$

The equation A6 can be converted to the following:

$$g_t^3(x_T - x_t) = \frac{\sqrt{\hat{V}_t}}{2\alpha(1-\beta_3)}[(x_t - x_T)^2 - (x_{t+1} - x_T)^2]+$$
$$\frac{\beta_3}{1-\beta_3}\frac{\hat{V}_t^{\frac{1}{4}}}{\sqrt{\alpha}}\frac{\sqrt{\alpha}S_{t-1}}{\hat{V}_t^{\frac{1}{4}}}(x_t - x_T) + \frac{\alpha}{1-\beta_3} \cdot \frac{\hat{S}^2}{\sqrt{\hat{V}_t}} \tag{A7}$$

Using Young's inequality ($ab \leq \frac{1}{2}(a^2 + b^2)$), we can infer:

$$g_t^3(x_T - x_t) \leq \frac{\sqrt{\hat{V}_t}}{2\alpha(1-\beta_3)}[(x_t - x_T)^2 - (x_{t+1} - x_T)^2]+$$
$$\frac{\beta_3}{2\alpha(1-\beta_3)}(x_t - x_T)^2\sqrt{\hat{V}_{t-1}} + \frac{\beta_3}{1-\beta_3}\frac{S_{t-1}^2}{\sqrt{\hat{V}_t}} + \frac{\alpha}{1-\beta_3} \cdot \frac{\hat{S}^2}{\sqrt{\hat{V}_t}} \tag{A8}$$

Inferring from **Lemma** 10.4 and **Theorem** 10.5 in Kingma's work and **Theorem** 4 in Reddi's work Reddi et al. (2019), using a sequence $\{1, 2, \cdots, T\}$ to replace $t$ in equation A8 to generate $t + 1$ equations, and calculate the summation of these equations, we have:

$$\Sigma_{t=1}^T g_t^3(x_t - x_T) \leq \Sigma_{i=1}^D \frac{1}{2\alpha(1 - \beta_3)}(x_1 - x_T)^2\sqrt{\hat{V}_{1,i}}+$$

$$\frac{1}{2(1 - \beta_3)}\Sigma_{i=1}^D \Sigma_{t=2}^T(\frac{\sqrt{\hat{V}_{t,i}}}{\alpha} - \frac{\sqrt{\hat{V}_{t-1,i}}}{\alpha}) + \Sigma_{i=1}^D \Sigma_{t=1}^T(x_t - x_t)^2\sqrt{\hat{V}_{t,i}} \tag{A9}$$

$$+K_3\Sigma_{i=1}^D \|g_{1:t,i}\|^2$$
$$K_3 < \infty$$

Inferring from **Theorem** 10.5 in Kigma's work Kingma & Ba (2014) and **Theorem** 4 in Reddi's work Reddi et al. (2019), we have:

$$\Sigma_{t=1}^T g_t^3(x_t - x_T) \leq \frac{K_1^2}{2\alpha(1 - \beta_3)}\Sigma_{i=1}^D\sqrt{T\hat{V}_{T,i}} + \frac{K_2}{2\alpha}\Sigma_{i=1}^D \Sigma_{t=1}^T \frac{\beta_{3,t}}{(1 - \beta_{3,t})}\sqrt{t\hat{V}_t}+$$

$$K_3\Sigma_{i=1}^D \|g_{1:t,i}\|^2 \tag{A10}$$
$$K_1, K_2, K_3 < \infty$$

Furthermore, we use a sequence $\{1, 2, \cdots, T - 1\}$ to replace $t$ in equation A10 and calculate the sum of these equations. According to **Assumption** 2.1, we can infer:

$$\Sigma_{t=1}^{T-1}(f(x_t) - f(x_T)) \leq \Sigma_{t=1}^{T-1}g_t \cdot (x_t - x_{t+1}) \tag{A11}$$

According to **Assumption** 2.3 and *Intermediate Value Theorem*, we have:

$$\Sigma_{t=1}^{T-1}g_t^3 \cdot (x_t - x_{t+1}) = \Sigma_{t=1}^T g_t^3 \cdot (x_t - x_T) = g^3 \tag{A12}$$

Inferring from equation A10 and equation A12, we conclude:

$$\|g\| \leq (\left\|\frac{K_1^2}{2\alpha(1 - \beta_3)}\Sigma_{i=1}^D\sqrt{T\hat{V}_{T,i}} + \frac{K_2^2}{2\alpha}\Sigma_{i=1}^D \Sigma_{t=1}^T \frac{\beta_{3,t}}{(1 - \beta_{3,t})}\sqrt{t\hat{V}_t} + K_3\Sigma_{i=1}^D \|g_{1:t,i}\|^2\right\|)^{\frac{1}{3}}$$

$$K_1, K_2, K_3 < \infty \tag{A13}$$

Inferring from equation A13, considering $T$ is sufficiently large, we have:

$$\|g\| = O(T^{1/6}) \tag{A14}$$

Let $\left\|\Sigma_{t=1}^{T-1}(f(x_t) - f(x_T))\right\|$ be $RES$. Inferring from equation A14 and **Assumption** 2.4, we have:

$$\frac{RES}{T} \leq \frac{\left\|\Sigma_{t=1}^{T-1}g_t \cdot (x_t - x_T)\right\|}{T} = \frac{\|\eta g\|}{T} = O(1/T^{5/6}) \tag{A15}$$

Finally, we conclude:

$$\frac{\|RES\|}{T} = O(\frac{1}{T^{\frac{5}{6}}}) \tag{A16}$$

It demonstrates the *HOME*-3 can reach to the convergence bound $O(\frac{1}{T^{\frac{5}{6}}})$ when $\|g_t - g\| < \epsilon, \forall \epsilon > 0$ and $\|g_t\|$ is sufficiently large. The following proof demonstrates that the convergence bound could be reduced when the gradient norm $\|g_t\|$ becomes smaller at the late stage.

2). On the other hand, we investigate the convergence bound when $\|g_t\| < 1$ for any $t$.

We can infer from **Assumption** 2.1 and equation A16. Then we have:

$$\frac{RES}{T} \leq \frac{K_1^2}{2\alpha(1 - \beta_3)}\Sigma_{i=1}^D\sqrt{T\hat{V}_{T,i}} + \frac{K_2^2}{2\alpha}\Sigma_{i=1}^D \Sigma_{t=1}^T \frac{\beta_{3,t}}{(1 - \beta_{3,t})}\sqrt{t\hat{V}_t}+$$

$$K_3\Sigma_{i=1}^D \|g_{1:t,i}\|^2 \tag{A17}$$
$$K_1, K_2, K_3 < \infty$$

Similarly, suppose $T$ is sufficiently large, we can conclude:

$$\frac{\|RES\|}{T} = O(\frac{1}{T^{\frac{1}{2}}}) \tag{A18}$$

We have proved **Theorem** 4.1. **Theorem** 4.1 demonstrate that *HOME*-3 can provide the convergence upper bound between $O(\frac{1}{T^{\frac{1}{2}}})$ and $O(\frac{1}{T^{\frac{5}{6}}})$. To summarize, the beginning gradient is usually large, *HOME*-3 provides a better convergence bound approximately to $O(\frac{1}{T^{\frac{5}{6}}})$. In the late stage, with the norm of gradient gradually reduced, the convergence bound of *HOME*-3 decreases to $O(\frac{1}{T^{\frac{1}{2}}})$. The performance of *HOME*-3 is comparable to Adam Kingma & Ba (2014) in the late stage, such as the gradient getting stuck in a stationary point.

**Theorem 4.2** Let $f$ satisfy **Assumption** 2, suppose $T$ as the maximum iteration, inferring from **Definitions** 3, 5, and 6, then $\frac{\|f(x_0)-f(x_T)\|}{T} = O(1/T^{5/6})$ holds.

***Proof***:
1) At the early stage, the norm of gradient $g_t$ is sufficiently large, and the higher-order momentum using $g_t^3$ dominates the update.

According to **Assumption** 2.2, we have:

$$f(x_{t+1}) - f(x_t) \le g_t(x_{t+1} - x_t) + \frac{L}{2}(x_{t+1} - x_t)(x_{t+1} - x_t)^T \tag{A19}$$

Since $(x_{t+1} - x_t)$ and $(x_{t+1} - x_t)^T$ are bounded, we let

$$\left\| (x_{t+1} - x_t)(x_{t+1} - x_t)^T \right\| \le K_M \left\| (x_{t+1} - x_t) \right\| \tag{A20}$$

Next, we use a sequence $\{1, 2, \cdots, T-1\}$ to replace $t$ in equation A16 and calculate the sum of these equations. We can infer:

$$\|f(x_1) - f(x_T)\| \le \left\| \Sigma_{t=1}^{T-1} g_t \cdot (x_{t+1} - x_t) + \frac{L}{2} \cdot (x_T - x_1) \right\| \tag{A21}$$

According to **Definition** 2.2, $L < \infty$, thus, $\|f(x_1) - f(x_T)\|$ only relates to term $\left\| \Sigma_{t=1}^{T-1} g_t \cdot (x_{t+1} - x_t) \right\|$.

Since $\left\| g_t^3 \right\| >> \|g_t\|, \forall t \in \{1, t\}$, we can infer:

$$\left\| \Sigma_{t=1}^{T-1} g_t \cdot (x_{t+1} - x_t) \right\| \le \left\| g_t^3 \right\| \cdot \left\| \Sigma_{t=1}^{T-1} (x_{t+1} - x_t) \right\| \tag{A22}$$

According to equation A20, equation A21, and equation A22 in **Theorem** 4.1, under **Assumption** 2.2, similarly, we can conclude:

$$\frac{\|f(x_1) - f(x_T)\|}{T} \le \frac{1}{T} \cdot \left\| \Sigma_{t=1}^{T-1} g_t \cdot (x_t - x_{t+1}) \right\| + \frac{K_M}{2T} \tag{A23}$$

Since we previously proved $\|g_t\| = O(T^{\frac{1}{6}})$, suppose $T$ is sufficiently large, we can infer $\frac{1}{T} \cdot \left\| \Sigma_{t=1}^{T-1} g_t \cdot (x_t - x_{t+1}) \right\|$ is equal to $O(\frac{1}{T^{\frac{5}{6}}})$.

Thus, *HOME*-3 can reach the convergence bound $O(\frac{1}{T^{\frac{5}{6}}})$ when the norm of gradient is sufficiently large.

On the other hand, considering the norm of gradient is not large. In that case, the lower-order momentum using $g_t$ can dominate the process.

Similar to equation A22 and equation A23, we can infer:

$$\frac{\|f(x_1) - f(x_T)\|}{T} \le \left\| \Sigma_{t=1}^{T-1} g_t \cdot (x_t - x_{t+1}) \right\| + \frac{K_M}{T} \tag{A24}$$

Since $\frac{1}{T} \cdot \left\| \Sigma_{t=1}^{T-1} g_t \cdot (x_t - x_{t+1}) \right\| = O(\frac{1}{T^{\frac{1}{2}}})$, we proved that *HOME*-3 can obtain convergence bound $O(\frac{1}{T^{\frac{1}{2}}})$ when the norm of gradient is not large.

In conclusion, *HOME*-3 can provide a comparable convergence bound under the smooth nonconvex Assumption (please refer to **Assumption** 2.2). The only potential issue is the smoothness of the objective function. If $L >> T$ in equation A21, the convergence bound could be seriously influenced.

**Lemma 4.3** (Norm of Coordinate Randomization Operator is Equal to 1) Suppose the permutation randomization as an operator $\mathcal{R} : \mathbb{R}^D \to \mathbb{R}^D$, $\|\mathcal{R}\| = 1$ holds, if $D < \infty$.

***Proof***:
Considering $\mathcal{R}$ applying on finite-dimensional space:

$$
\mathcal{R} \cdot \begin{bmatrix} x_1 \\ x_2 \\ \vdots \\ x_D \end{bmatrix} = \begin{bmatrix} \hat{x}_1 \\ \hat{x}_2 \\ \vdots \\ \hat{x}_D \end{bmatrix} \tag{A25}
$$

Inferring from equation A13, we have:

$$
\hat{x}_1 = x_i, \hat{x}_2 = x_j, \cdots, \hat{x}_D = x_k, i, j, k \in [1, D] \tag{A26}
$$

Inferring from equation A26, we have:

$$
\|\{x_1, x_2, \cdots, x_D\}\| = \|\{\hat{x}_1, \hat{x}_2, \cdots, \hat{x}_D\}\| \tag{A27}
$$

According to the concept of operator norm (Rudin, 1973), we can derive the following:

$$
\|\mathcal{R}\| = sup \frac{\mathcal{R} \cdot \|\{x_1, x_2, \cdots, x_D\}\|}{\|\{x_1, x_2, \cdots, x_D\}\|} = sup \frac{\|\{\hat{x}_1, \hat{x}_2, \cdots, \hat{x}_D\}\|}{\|\{x_1, x_2, \cdots, x_D\}\|} = 1 \tag{A28}
$$

**Theorem 4.4** (Coordinate Randomization Maintains The Convergence Bound of Incorporated Optimizer) Inferring from **Lemma** 4.3, the convergence bound of a gradient-based optimizer incorporating coordinate randomization $\mathcal{R} \cdot \mathcal{G}$ should be equal to the convergence bound of an original gradient-based optimizer $\mathcal{G}$ without coordinate randomization.

***Proof***:
Inferring from the concept of contraction operator, we have:

$$
\|\mathcal{G} \cdot (f(X) - f(Y))\| \le c\|\mathcal{G} \cdot (f(X) - f(Y))\| \\
0 < c < 1 \tag{A29}
$$

We can rewrite the left side of equation A16 as:

$$
\|\mathcal{G} \cdot (f(I_{t+1}) - f(I_t))\| \tag{A30}
$$

Then, we have:

$$
\|\mathcal{G} \cdot (f(I_{t+1}) - f(I_t))\| \le c \cdot \|(f(I_{t+1}) - f(I_t))\| \tag{A31}
$$

Considering the incorporation of optimizer and randomization as $\mathcal{R} \cdot \mathcal{G} \cdot f(x)$, we have

$$
\|\mathcal{R} \cdot \mathcal{G} \cdot (f(I_{t+1}) - f(I_t))\| \le \|\mathcal{R}\| \cdot \|\mathcal{G} \cdot (f(I_{t+1}) - f(I_t))\| \tag{A32}
$$

Inferring from **Lemma** 4.3, it is obvious that we have:

$$
\|\mathcal{R}\| \cdot \| \cdot \mathcal{G} \cdot (f(I_{t+1}) - f(I_t))\| = \|\mathcal{G} \cdot (f(I_{t+1}) - f(I_t))\| \le c \cdot \|f(I_{t+1}) - f(I_t)\| \tag{A33}
$$

equation A33 implies permutation randomization $\mathcal{R}$ can maintain the convergence rate of original gradient-based optimizer $\mathcal{G}$.

**Additional Experiments**:

In additional experiments, we compare the time consumption of *HOME*-3 with other peer optimizers.

Table 2: Time Consumption Comparison in Seconds of *HOME*-3 and Other Peer Three Optimizers

| Time Consumption at 1st Layer | Time Consumption at 2nd Layer |
|---|---|
| ADMM $431.58 \pm 83.56$ | ADMM $247.42 \pm 68.54$ |
| ADAM $961.65 \pm 199.67$ | ADAM $585.37 \pm 55.17$ |
| STORM $4711.35 \pm 342.25$ | STORM $4616.66 \pm 556.27$ |
| HOME-3 $1262.66 \pm 195.16$ | HOME-3 $1108.62 \pm 188.05$ |

Moreover, to ensure a fair comparison among different methods for optimizing supervised learning problems, we set all parameters to reported default values Kingma & Ba (2014); Cutkosky & Orabona (2019). Each method was then employed to solve a logistic regression problem (Schober & Vetter, 2021) using publicly released breast cancer data Shut (2023) for classification. The results, observed within iterations 1 to 200, are illustrated in Figure 6.

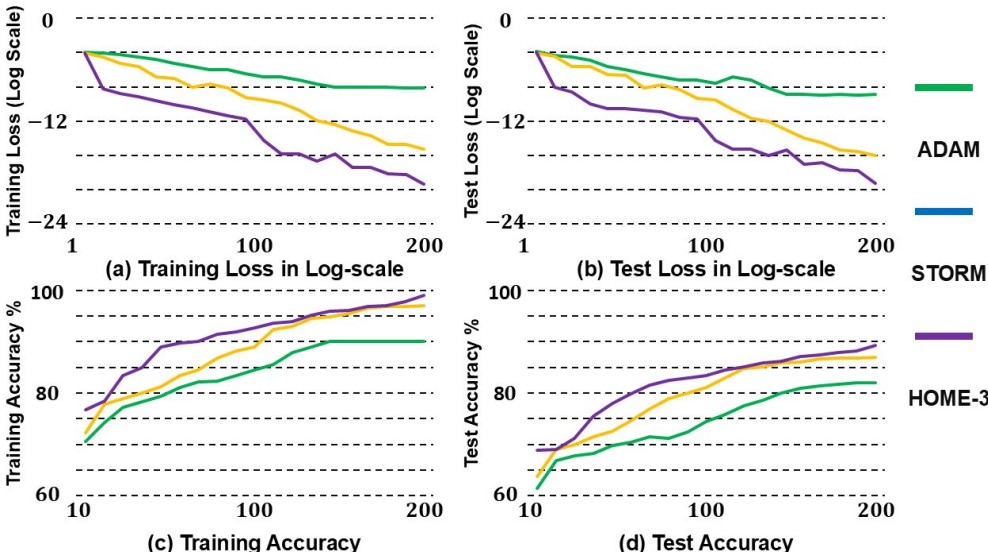

Figure 6: An illustration of reconstruction loss comparisons of *HOME*-3 and other peer optimizers on solving logistic regression problem.

