# OpenReview forum: "HOME-3: HIGH-ORDER MOMENTUM ESTIMATOR USING THIRD-POWER GRADIENT FOR CONVEX, SMOOTH NONCONVEX, AND NONSMOOTH NONCONVEX OPTIMIZATION"
_ICLR.cc/2025/Conference — ICLR 2025 Conference Withdrawn Submission_

### Official Review · Reviewer_wPJy · 2024-10-16

**Soundness:** 2
**Presentation:** 1
**Contribution:** 2
**Rating:** 3
**Confidence:** 4

**Summary:**

This paper proposes to leverage the third order momentum of gradients for optimization algorithms. Home-3 is proposed and its convergence is proved under different settings including convex, nonconvex smooth and nonconvex nonsmooth ones. Numerical results are also provided for dictionary learning, deep matrix factorization, and some neural networks.

**Strengths:**

The idea of employing third order momentum is interesting, and as far as I know, it is not explored in the literature.

**Weaknesses:**

**W1.** The paper is not well written, and sometimes to an extent that brings difficulty to evaluate the correctness of results.

- It is not clear what does assumption 1 refer to in theorem 4.1, and there is no assumption indexed with 1.
- Similarly, what is exactly assumption 2? It is hard to decipher from the context.
- The major theorems are presented in a way such that even the authors seem not too confident about. For example, in theorem 1, the authors wrote "..(some term)... can reach ...". It seems to be a statement that is very ambiguous -- only in the best case it holds. Similarly, "... should be equal to ..." in theorem 4 is also very confusing.

**W2.** The theoretical results need to be further discussed.

- Assumption 2.4 is too strong, and clearly $g_t$ depends on $g_{t-1}$, i.e., their correlation is not 0.
- The convergence criterion in this paper is $\|f(x_t) - f(x_T)\|$, which is quite different from commonly adopted suboptimality error or gradient norm. More importantly, it is not clear whether $\|f(x_t) - f(x_T)\|$ make sense. This term can be $0$ if we do not update at all, i.e., set $x_0= x_1=...=x_T$. In other words, even if this term is small, it is not straightforward to say anything about optimality. Lastly, when $\|f(x_0) - f(x_T)\|$ is small, it simply means that the algorithm does not make too much progress -- suppose that f(\x_0) is large, it means that f(x_T) is also large.
- The results here are not directly comparable to e.g., Adam or Storm, because of different metrics.

**W3.** The algorithm is not intuitively clear. See my questions below.

**Questions:**

**Q1.** Can the authors help me on the intuitive understanding of HOME3? In particular, if we ignore the momentum in S, M, V (by setting betas = 0), and consider the following settings respectively:
- large gradient regime, where $g_t$ is large entrywisely. In this case we have $g_t^3 \gg g_t$, which means that $M_t - S_t \approx -g_t^3$. And note that $[g_t]^i * [g_t^3]^i  > 0$, in other words, negative $M_t - S_t$ is not a descent direction!
- small gradient regime, where $g_t$ is small entrywisely. In this case we have $g_t^3 \ll g_t$, $M_t - S_t \approx g_t$, and the algorithm reduces to Adam. Then it does not explain the large performance gap between HOME3 and adam in the numerical results.

---

### Official Review · Reviewer_1cLF · 2024-10-20

**Soundness:** 2
**Presentation:** 1
**Contribution:** 1
**Rating:** 3
**Confidence:** 5

**Summary:**

This paper presents HOME-3 (High-Order Momentum Estimator using 3rd power gradient) optimizer for convex/non-convex optimization and its convergence analyses. It also provides some numerical results indicating that HOME-3 performs better than the existing optimizers.

**Strengths:**

The strength of this paper is to present HOME-3 (High-Order Momentum Estimator using 3rd power gradient) optimizer for convex/non-convex optimization. The originality of this paper is to apply 3rd power gradients to solve convex/non-convex optimization problems. Moreover, the significance of this paper is to provide numerical results showing that  HOME-3 performs well.

**Weaknesses:**

The weaknesses of this paper are:
- the mathematical preliminaries are poor.
- the proofs of the convergence analyses are incorrect.

**Questions:**

The following are my questions and suggestions:
- L103: $X \in \mathbb{R}^D \to X \subset \mathbb{R}^D$ (since $X$ is a set)
- L104: Is $D < \infty$ (i.e., $D \in \mathbb{R}$) correct? I think that $D \in \mathbb{N}$ is correct (since $D$ should not be negative).
- L112: $(\nabla f (x))^n$ is not defined. I guess $\nabla f (x) := ( \frac{\partial f (x)}{\partial x_1}, \cdots, \frac{\partial f (x)}{\partial x_D})^\top$ and $(\nabla f (x))^n := ( (\frac{\partial f (x)}{\partial x_1})^n, \cdots, (\frac{\partial f (x)}{\partial x_D})^n)^\top$. I think that $n < \infty$ should be replaced with $n \in \mathbb{N}$ (since $n$ should not be negative).
- L115: $\nabla^k f(x)$ is not defined.
- L120: suppose $t (\forall t \in \mathbb{N}) \to $ suppose $t \in \mathbb{N}$?
- Definition 2.4: I do not think that the definition is well-defined. Do you mean that $\mathcal{R}[x_1, \cdots, x_D]^\top = [\hat{x}_1, \cdots, \hat{x}_D]^\top$? $\hat{x}_i$ $(i=1,2,\cdots, D)$ is not defined.
- Definition 2.5: I do not think that the definition is well-defined. From $\mathcal{G} \colon \mathbb{R} \to \mathbb{R}^D$ and $f(x) \in \mathbb{R}$, $\mathcal{G} (f(x)) \in \mathbb{R}^D$ is well-defined. However, $\mathcal{G}^2 (f(x)) := \mathcal{G} [\mathcal{G} (f(x))]$ is not defined, since the domain of $\mathcal{G}$ is $\mathbb{R}$.
- Definition 2.6: I do not think $x_T$ is just a stationary point. Do you mean that $x_T$ is an approximated stationary point?
- Definition 2.7: I do not think that the definition is well-defined from the issue of Definition 2.5.
- Assumption 2.1: $\forall x, y \in \mathbb{R}^D, f(y) \geq ...$ is correct.
- Assumption 2.2: $\exists L > 0 \forall x, y \in \mathbb{R}^D, f(y) \leq ...$ is correct.
- Assumption 2.4: The definition indicates that $\forall T \forall n \in \mathbb{N} \forall \epsilon > 0$ $(\forall t \in [1,T] \to \exists [k_1, \cdots, k_T]^\top \in \mathbb{R}^T$ (2) holds). I do not know why Assumption 2.2 implies Assumption 2.4 (L155-L156). Could you prove that Assumption 2.2 implies Assumption 2.4?
- (3): Please add more explanations such that (3) is useful to solve optimization problems.
- (3): I think that the definition of $\sqrt{\hat{V}_{t-1}}$ (the square root of a vector) is needed.
- L206-L215: Does "the Eq. 3 equal to 0" imply that $x_t = x_{t-1}$? I cannot understand why (5) and steps 10-12 in Algorithm 1 are needed. In particular, the authors should explain the reasons why using (5) and steps 10-12 in Algorithm 1 escapes potential stationary points (not "potential" but "poor" is correct?).
- Table 1: $M_0, V_0, S_0$ are not defined. From Page 14, I guess that $M_0, V_0, S_0 = 0$.
- Theorem 4.1: There is no Assumption 1 in this paper. Write all of the assumptions and conditions used in the theorem. The Appendix section uses $\alpha_t = \alpha$ and $\epsilon_1 = 0$, while Table 1 uses $\alpha_t$ and $\epsilon_1 > 0$. $\Vert \sum ... \Vert$ is replaced with $| ... |$.
- L265: I do not understand what $\frac{L}{\sqrt{T}} \to 0, \forall x, y \in X$ is.

Proof of Theorem 4.1:
- (A2) does not hold, since $x_{t+1} \in \mathbb{R}^D$. Please use $\Vert x_{t+1} - x_T \Vert^2 = \Vert x_{t} - x_T \Vert^2 - \cdots$.
- It is not obvious that (A3) holds. To show (A3), the authors seem to use Assumption 2.4. Since Assumption 2.4 is an $\epsilon$-approximation, Assumption 2.4 does not lead to (A3). Please provide the proof of (A3).
- L724: It is not guaranteed that $\hat{S}_t$ is bounded. Also, (A4) does not imply that $\hat{S}_t$ is bounded, since it is a possibility such that $\Vert g_T \Vert \to \infty$ when $T \to \infty$. Please prove the boundedness of $\hat{S}_t$.
- L728, L729: Please also prove the boundedness of $\hat{M}_t$ and $\hat{V}_t$.
- (A5): The authors use $\hat{M}_t = \frac{M_t}{1 - \beta_1^t}= 0$ to show (A5). It is incorrect. If the authors could use $\hat{M}_t = 0$, then the following steps are needed.

Case 2): Suppose that $\forall t$ $\Vert g_t \Vert < 1$ (See Page 15).

Case 1): Suppose that $\lnot(\forall t \text{ }\Vert g_t \Vert < 1) = \exists t_0 \text{ } \Vert g_{t_0} \Vert \geq 1$ (See also L731; L731 is not correct).
- Case 1-1): Suppose that $\hat{M}_{t_0} = 0$.
- Case 1-2): Suppose that $\hat{M}_{t_0} \neq 0$.

Therefore, I strongly believe that the current proof of Theorem 4.1 is incorrect.

- Theorem 4.2: The proof of Theorem 4.2 is based on Theorem 4.1. Moreover, Theorem 4.4 is based on $\mathcal{G}^t$, which is not well-defined (See my comment for Definition 2.5). Therefore, Theorems 4.2 and 4.3 are also incorrect mathematically.

---

### Official Review · Reviewer_QSEs · 2024-10-28

**Soundness:** 1
**Presentation:** 1
**Contribution:** 2
**Rating:** 3
**Confidence:** 2

**Summary:**

This paper explores advancements in gradient-based optimization by introducing a novel High-Order Momentum Estimator (HOME) optimizer, focusing on the use of third-order momentum to improve convergence rates. The key idea is to extend traditional momentum techniques, which primarily use first- and second-order gradients, to incorporate higher-order gradients and accelerate convergence, especially in challenging optimization landscapes like nonsmooth nonconvex problems.

The proposed HOME-3 optimizer is accompanied by faster convergence rates for both convex and smooth nonconvex problems. The authors also empirically validate the effectiveness of HOME-3 on nonsmooth nonconvex problems, including deep neural networks, where it reportedly outperforms existing momentum-based optimizers like Adam and STORM.

**Strengths:**

Incorporating higher-order momentum into existing optimization algorithms is an interesting direction that can lead to accelerated rates.

**Weaknesses:**

- Section 2 is somewhat difficult to read without accompanying intuition or explanatory text. Adding a brief development or context for these definitions could improve clarity. In addition, some definitions are really unclear; e.g., in Definition 2.4, what does it mean for $\mathcal{R}$ to be a ''coordinate randomization''?

- It is hard to follow the formulas when the terms have not yet been defined. For example, in the core algorithmic update of Equation (3), $\hat M_t, \hat V_t, \hat S_t$ are the ''first-order, second-order, and third-order momentum'' and the authors refer to Definition 2.1, but the latter never defined these. The definitions are only presented later within Algorithm 1.

- Many of the formulas lack clarity. For example, in Equation (3), there is a division by $\hat V_t$, which is supposedly a vector.

- Quantities like $\epsilon_1$ should be explicitly specified, rather than just ''defined the same as in Adam''.

Due to significant ambiguities in the definitions and the lack of clarity for key components of the algorithm, as mentioned above, I found it difficult to fully understand the proposed method. As a result, it’s challenging to assess the validity or implications of the theoretical and experimental results presented in later sections.

**Questions:**

Aside from the weaknesses above,
- The paper makes multiple uses of quantities like $(\nabla f(x))^n$ (e.g., Definition 2.1). Given that $\nabla f(x)$ is a vector, what exactly does this mean?

- From the descriptions, it seems that $\hat V_t$ is a vector. Then, what does it mean to divide by it in Equation (3)? How about adding the scalar $\epsilon_1$ to it?

---

### Official Review · Reviewer_2FhE · 2024-10-30

**Soundness:** 2
**Presentation:** 2
**Contribution:** 2
**Rating:** 5
**Confidence:** 2

**Summary:**

This paper proposes a deterministic first-order algorithm called HOME-3 for convex, smooth and non-smooth nonconvex optimization, leveraging the idea of Adam while introducing an extra momentum term with the third power of the gradient norm. The authors provide the convergence analysis for convex and smooth nonconvex optimization and claim that this algorithm achieve a convergence rate of O(1/T^(5/6)) for finding an approximation stationary point. For non-smooth nonconvex optimization, the authors conduct some experiments to show the effectiveness of the proposed algorithm.

**Strengths:**

This paper proposes a deterministic first-order algorithm called HOME-3 for convex, smooth and non-smooth nonconvex optimization, leveraging the idea of Adam while introducing an extra momentum term with the third power of the gradient norm.  I think the main strength is that the proposed HOME-3 algorithm introduces an extra momentum term rather than two momentum terms of Adam.

**Weaknesses:**

1. The overall presentation is not clear. Specifically, in section 2 where some definitions and assumptions are given as preliminaries and section 4 where the convergence results of the proposed algorithm are analyzed, the description is very dense and difficult to parse.

2. About Definition 2.2. The authors give an unusual definition that any kth order derivative of the objection function f with k∈N is Lipschitz continuous with a fixed Lipschitz constant L, without further description, which is not make sense.

3. Another major weak point in the presentation is a lack of clear comparison with prior work. The authors mentioned a few previous algorithms such as Adam, STORM and STORM+ and their convergence rate. However, it’s not clear that whether these results are comparable, since I’m not sure if they have the same definition of smoothness and use Assumption 2.3 and 2.4.

4. Although the paper claim that the proposed HOME-3 achieve a convergence rate of O(1/T^(5/6)) for finding an approximation stationary point, I’m not sure whether this result is reasonable. According to Carmon et al. [1], for a class of nonconvex function with L_p-Lipschitz continuous derivatives for all q∈{1,...,p}, where p∈N, there does not exist a deterministic first-order algorithm with iteration complexity better than O(ϵ^(-8/5)), which is contradictory to the result proposed in this paper.

5. In section 3 the authors give a stopping criterion ‖M ̂_t-S ̂_t ‖<ϵ but lack of analysis about the relationship between this criterion and the common definition of stationarity in nonconvex optimization.

6. Though the paper leverages the idea of Adam, the proposed algorithm is only analyzed in deterministic setting. Why not give a further convergence analysis in stochastic smooth convex/nonconvex optimization?

7. Regarding the experiments, the authors conduct several small-scale tasks with 100 or 200 iterations. Addition large-scale experiments should be conduct considering the analysis of the proposed algorithm in smooth nonconvex case is based on T is sufficiently large according to section 4.2.

[1] Carmon Y, Duchi J C, Hinder O, et al. Lower bounds for finding stationary points II: first-order methods[J]. Mathematical Programming, 2021, 185(1): 315-355.

**Questions:**

1. The overall presentation is not clear. Specifically, in section 2 where some definitions and assumptions are given as preliminaries and section 4 where the convergence results of the proposed algorithm are analyzed, the description is very dense and difficult to parse.

2. About Definition 2.2. The authors give an unusual definition that any kth order derivative of the objection function f with k∈N is Lipschitz continuous with a fixed Lipschitz constant L, without further description, which is not make sense.

3. Another major weak point in the presentation is a lack of clear comparison with prior work. The authors mentioned a few previous algorithms such as Adam, STORM and STORM+ and their convergence rate. However, it’s not clear that whether these results are comparable, since I’m not sure if they have the same definition of smoothness and use Assumption 2.3 and 2.4.

4. Although the paper claim that the proposed HOME-3 achieve a convergence rate of O(1/T^(5/6)) for finding an approximation stationary point, I’m not sure whether this result is reasonable. According to Carmon et al. [1], for a class of nonconvex function with L_p-Lipschitz continuous derivatives for all q∈{1,...,p}, where p∈N, there does not exist a deterministic first-order algorithm with iteration complexity better than O(ϵ^(-8/5)), which is contradictory to the result proposed in this paper.

5. In section 3 the authors give a stopping criterion ‖M ̂_t-S ̂_t ‖<ϵ but lack of analysis about the relationship between this criterion and the common definition of stationarity in nonconvex optimization.

6. Though the paper leverages the idea of Adam, the proposed algorithm is only analyzed in deterministic setting. Why not give a further convergence analysis in stochastic smooth convex/nonconvex optimization?

7. Regarding the experiments, the authors conduct several small-scale tasks with 100 or 200 iterations. Addition large-scale experiments should be conduct considering the analysis of the proposed algorithm in smooth nonconvex case is based on T is sufficiently large according to section 4.2.

[1] Carmon Y, Duchi J C, Hinder O, et al. Lower bounds for finding stationary points II: first-order methods[J]. Mathematical Programming, 2021, 185(1): 315-355.

---

### Note · Authors · 2024-11-12

I have read and agree with the venue's withdrawal policy on behalf of myself and my co-authors.